# Assessing the influence of the modifiable areal unit problem on Bayesian disease mapping in Queensland, Australia

**Farzana Jahan**[1,2☯]*, **Shovanur Haque**[2☯], **James Hogg**[2,3], **Aiden Price**[2,3], **Conor Hassan**[2,3], **Wala Areed**[2,3], **Helen Thompson**[2,3], **Jessica Cameron**[5☯], **Susanna M. Cramb**[2,4☯]

**1** School of Mathematics, Statistics, Chemistry & Physics, College of Science, Technology, Engineering and Mathematics, Centre for Healthy Ageing, Health Futures Institute, Murdoch University, Perth, Western Australia, Australia, **2** QUT Centre for Data Science, Queensland University of Technology, Brisbane, Queensland, Australia, **3** School of Mathematical Sciences, Queensland University of Technology, Brisbane, Queensland, Australia, **4** Australian Centre for Health Services Innovation, School of Public Health and Social Work, QUT, Brisbane, Queensland, Australia, **5** Descriptive Epidemiology, Cancer Council Queensland (CCQ), Brisbane, Queensland, Australia

☯ These authors contributed equally to this work.
* fjahan0518@gmail.com

**Data Availability Statement:** Simulated lung cancer data for Queensland used in the research are made available in the form of supplmentary information.

## Abstract

### Background

Spatial data are often aggregated by area to protect the confidentiality of individuals and aid the calculation of pertinent risks and rates. However, the analysis of spatially aggregated data is susceptible to the modifiable areal unit problem (MAUP), which arises when inference varies with boundary or aggregation changes. While the impact of the MAUP has been examined previously, typically these studies have focused on well-populated areas. Understanding how the MAUP behaves when data are sparse is particularly important for countries with less populated areas, such as Australia. This study aims to assess different geographical regions' vulnerability to the MAUP when data are relatively sparse to inform researchers' choice of aggregation level for fitting spatial models.

### Methods

To understand the impact of the MAUP in Queensland, Australia, the present study investigates inference from simulated lung cancer incidence data using the five levels of spatial aggregation defined by the Australian Statistical Geography Standard. To this end, Bayesian spatial BYM models with and without covariates were fitted.

### Results and conclusion

The MAUP impacted inference in the analysis of cancer counts for data aggregated to coarsest areal structures. However, area structures with moderate resolution were not

**Funding:** QUT Centre for Data Science.

**Competing interests:** NO authors have competing interests.

greatly impacted by the MAUP, and offer advantages in terms of data sparsity, computational intensity and availability of data sets.

## Introduction

It is well known that health outcomes vary by residential location. Spatial modelling of health data (e.g. disease mapping) can provide critical insights into the geographic patterns of and the relationships between health outcomes and their determinants. Aggregating spatial data by area is a popular practice in disease mapping as it helps: (1) identify clustering of disease, (2) provide insights into the spatial variability of disease burden, and (3) identify health disparities across different geographic regions [1]. Spatial aggregation can also help protect confidentiality and aid in calculation of pertinent rates and risks. But aggregated spatial data are susceptible to the modifiable areal unit problem (MAUP), where the choice of spatial units used to aggregate the data can significantly impact the analysis results.

The term, MAUP, was used for the first time by Openshaw and Taylor [2] showing the systematic variation in correlation values using different boundary systems for analysis. The MAUP may emerge from two different mechanisms [3]: the zoning effect, where the total number of areas is kept constant but boundaries are redefined; and the scaling effect, where the spatial resolution or aggregation level is altered leading to changes in the total number of areas. Analysing datasets from the same population using different zonations or scales may lead to inconsistent results, which is the essence of the MAUP [3]. Multiple cancer studies have demonstrated that different methods of aggregating health data led to varying interpretations of disease clusters (e.g. [4, 5]), with different zoning schemes and scales of aggregation resulting in different clusters being identified and impacting conclusions about high-risk areas.

Some have argued that aggregation beyond the minimal level yields untrustworthy results [6], while others believe there can be equally valid but different conclusions from analysing data at different aggregation levels [7]. Understanding the MAUP is crucial for analysis of spatial data at the small area level [8], since there are still several unresolved issues [9].

Ongoing research is examining how the MAUP affects various types of data and different geographical structures [7, 10–13], however, Australia's geography is unique due to its vast geographical size, diverse ecological systems, and highly heterogeneous population distribution. As a result, the MAUP may have different implications for health data analysis in Australia compared to other countries. To date, there are only a handful of published works [6, 11, 14] investigating the impact of the MAUP in the Australian context.

In Australia, MAUP studies have modelled emergency department [11] or hospital data [14] using the popular Bayesian Besag, York and Mollié (BYM) model [15] in the city of Perth, Western Australia. There were 3890 SA1s and 164 SA2s in Perth in 2011, considered in this study, with median populations of 400 (for statistical areas level 1 (SA1s)), and 10000 (for larger SA2s). These studies have proposed various solutions to the MAUP, including using an overlay aggregation method for disease mapping by incorporating information from two nested aggregate levels [14], or minimising the influence of the MAUP through only reporting results at very high resolution, such as SA1s [11]. However, there are far greater numbers of people who present to hospital emergency departments or are admitted to hospitals than experience less common diseases or conditions.

Internationally, there has been some examination of MAUP for less common diseases. In the United States MAUP was evaluated when modelling breast cancer incidence in Indiana

State Cancer Registry from 2010 to 2015, but even at the smallest levels examined, areas had between 600 to 3000 residents [16]. Their suggestion to avoid MAUP was to disaggregate counts to approximate individual-point locations, as occurred for a New Hampshire study on birth defects [17]. This used Restricted and Controlled Monte Carlo (RCMC) to disaggregate area-level counts by randomly placing individuals in pixels within the original area, with a higher probability of being placed where the population was higher. Disadvantages included running multiple times to account for the inherent uncertainty in placement, and then being restricted to using analyses designed for individual data.

Investigating the impact of the MAUP in regions with sparse data is crucial for accurate and reliable spatial analyses. In countries like Australia, where large areas have low population densities, this investigation is particularly important to inform policies and decisions that are equitable, effective, and tailored to the unique characteristics of these regions. Understanding the MAUP ensures that data-driven decisions truly reflect the realities of these areas, leading to better outcomes for their inhabitants and environments.

Given the limited suggestions offered to date, there remains a significant opportunity to elucidate the influence of the MAUP within sparse data contexts. The impact of the MAUP has not previously been explored on outcomes with low counts and very sparse populations, such as cancer within rural and remote Australia. The present study aims to fill this research gap by exploring the implications of Bayesian spatial modelling applied to sparse data aggregated at diverse boundary levels across an entire state in Australia.

In this paper we explore the impact of the MAUP using Bayesian spatial models applied to synthetic-style cancer data, which has lower overall counts than previous studies, across different levels of aggregation in Queensland, Australia. We consider the impact on both spatial disease patterns and covariate inference.

## Materials and methods

### Spatial structures

The Australian Statistical Geography Standard (ASGS) 2016, developed by the Australian Bureau of Statistics, defines a hierarchy of area structures with five levels of resolution. The area structure with the finest resolution is termed "mesh blocks", and the state of Queensland has 67,047 mesh blocks with a median population of 82. The remaining levels of the hierarchy are termed "Statistical Areas" (SA) and the resolution decreases from SA1 (n = 11,507 areas, median population of 401) to SA4 (n = 19, median population of 198,975), excluding the zero population areas. The hierarchy is nested, so that each mesh block lies wholly within a single SA1, each SA1 lies within a single SA2 and so on.

### Data

Simulated lung cancer incidence data for Queensland, Australia were provided by Cancer Council Queensland and reflects the distribution for lung cancer incidence over a ten year period, 2005–2014 [18]. The lung cancer incidence (number of cases) were simulated at the SA2 level, and then aggregated to SA3 and SA4 levels, and dis-aggregated to SA1 and mesh block levels using publicly available geographical correspondence files [19]. The correspondence files enable more accurate aggregations and dis-aggregations by accounting for the total populations of mesh and SA levels [20]. Counts and populations at each level of aggregation were calculated by adding the counts and populations for the respective areas. At each level of aggregation, indirectly-standardised incidence rates for area $i$ ($SIR_{obs,i}$) were calculated as the ratio of the observed counts ($y_i$) divided by the expected counts ($E_i$), based on the state-wide observed counts in age group $k$ ($y_k$), and the population in age group $k$ for area $i$ ($pop_{ik}$) and

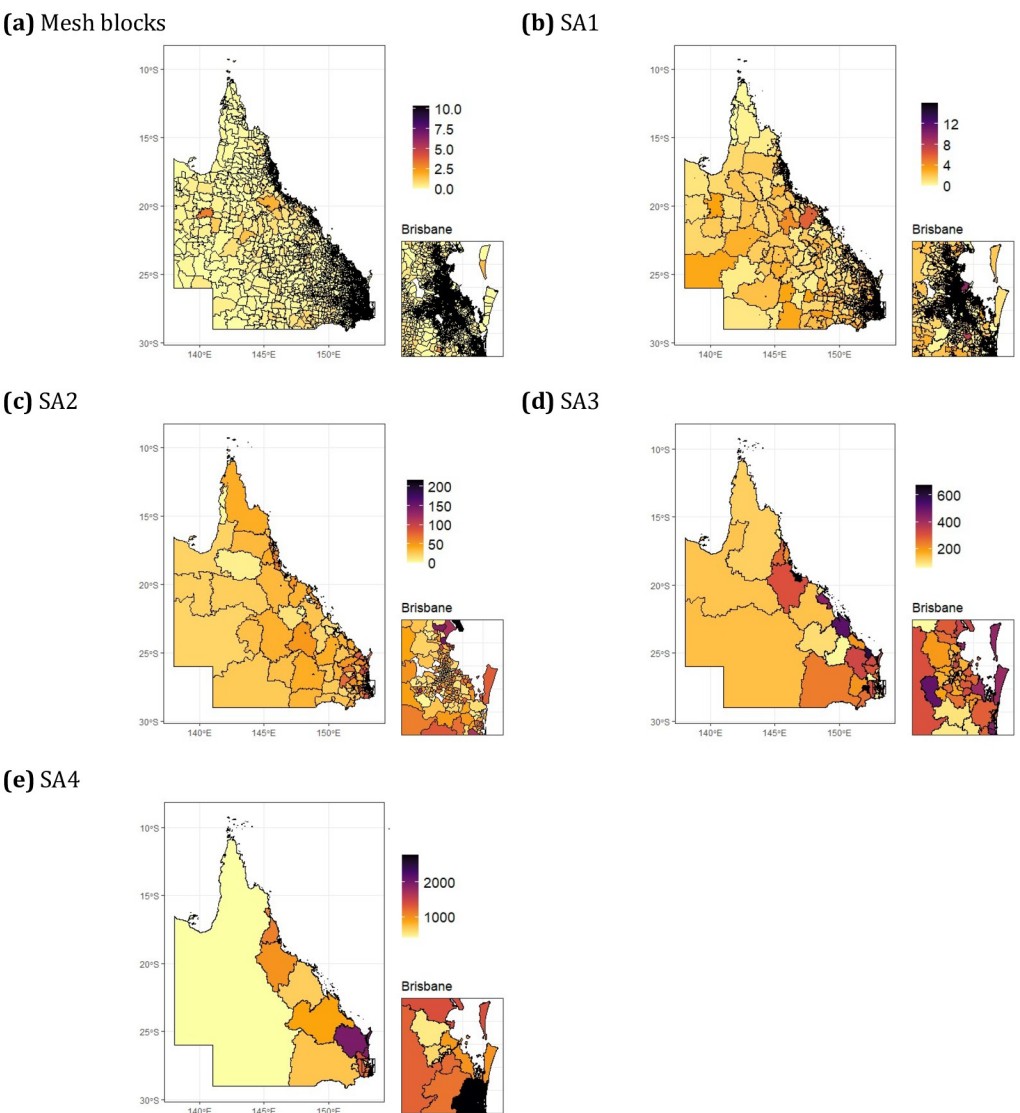

**Fig 1. Choropleth maps displaying the simulated counts of lung cancer at multiple aggregation levels for Queensland, Australia.** Note that the maps for each aggregation level have different colour scales to reflect how the average counts per area increases as aggregation occurs.

state-wide ($pop_k$), as follows.

$$SIR_{obs,i} = y_i/E_i \text{ for } i = 1, 2, \cdots, Nareas,$$

$$E_i = \sum_{k=1}^{K} \frac{y_k pop_{ik}}{pop_k}.$$

Fig 1 shows the simulated cancer counts for each of the different aggregation levels in Queensland, followed by the observed standardised incidence rates in Fig 2. Disease counts are not a good representation of disease risk, since they do not account for the total population in the area. Hence the SIR, which is the ratio of observed to expected incidence cases, is used to

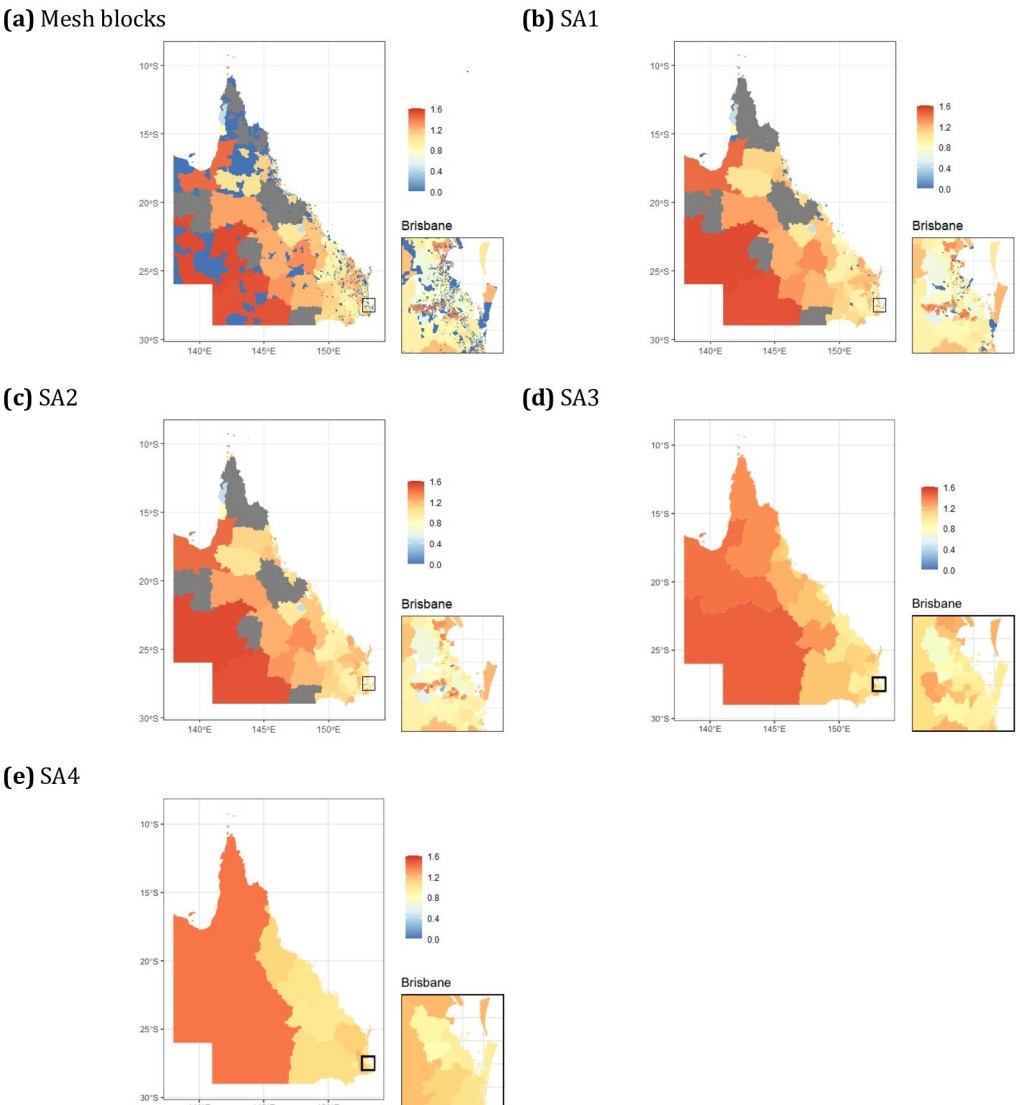

**Fig 2. Choropleth maps displaying the observed SIR from the simulated lung cancer data at multiple aggregation levels for Queensland, Australia (grey regions have 0 cases).**

summarise the underlying pattern of lung cancer incidence in the simulated Queensland data. Table 1 presents the descriptive statistics of the data.

**Socio-Economic Indexes for Areas (SEIFA).** Lung cancer is known to have a strong association with socioeconomic factors, with higher incidence among areas with greater socioeconomic disadvantage [21]. To study the impact of the MAUP on inference at different levels of spatial aggregation, we included a covariate: area-level socioeconomic disadvantage. We used the Australian Bureau of Statistics' Index of Relative Socioeconomic Disadvantage (IRSD), categorised into quintiles, where a region in the first quintile is amongst the most disadvantaged regions and a region in the fifth quintile is amongst the least disadvantaged regions [22]. The IRSD scores and quintiles are publicly available at the SA1 and SA2 level, and thus, SA2-level IRSD scores were disaggregated to mesh block and aggregated to higher regions (SA3 and

**Table 1. Population and number of lung cancer diagnoses by geographic levels of aggregation.**

| Aggregate level | Number of areas | Mean | SD | Min | Median | IQR | Max |
|---|---|---|---|---|---|---|---|
| | | | Population | | | | |
| Mesh | 67047.0 | 87.0 | 57.7 | 3.0 | 82 | 63.0 | 2446.0 |
| SA1 | 11507.0 | 422.0 | 181.4 | 14.0 | 401.0 | 186.0 | 5156.0 |
| SA2 | 507.0 | 7737.0 | 3932.6 | 7.0 | 7120.0 | 4764.0 | 23338.0 |
| SA3 | 82.0 | 48878.0 | 24660.5 | 10487.0 | 45363.0 | 22764 | 159519.0 |
| SA4 | 19.0 | 160431.0 | 83677.4 | 81082.0 | 198975.0 | 91187 | 462573.0 |
| | | | Lung cancer diagnosis | | | | |
| Mesh | 67047.0 | 0.3 | 0.3 | 0 | 0.3 | 0.4 | 10.0 |
| SA1 | 11507.0 | 1.9 | 1.1 | 0 | 1.7 | 1.3 | 16 |
| SA2 | 507.0 | 39.6 | 27.1 | 0 | 33.0 | 28 | 216.0 |
| SA3 | 82.0 | 252.4 | 134.2 | 55.0 | 223.4 | 151.3 | 672.0 |
| SA4 | 19.0 | 1089.4 | 590.5 | 396.2 | 972.0 | 573.6 | 2778 |

SD stands for standard deviation, IQR stands interquartile range is calculated
as the difference between 75$^{th}$ and 25$^{th}$ percentile

SA4) following the methods recommended by the Australian Bureau of Statistics [23], and formed into quintiles.

**Ethical approval.** Only simulated data were used in this manuscript, lung cancer data was simulated by Cancer Council Queensland following the appropriate ethical consideration [18]. This was approved by the Data Custodians of the original data. The other datasets used in this manuscript are the geographical boundary levels, population counts which are publicly available via Australian Bureau of Statistics website [24].

## Statistical methods

The impact of the MAUP on varying area levels was assessed by investigating the underlying spatial autocorrelation in observed data and model residuals. We also compared the posterior summaries of the parameters of interest from the Bayesian spatial models fitted to five different aggregate levels (mesh blocks to SA4 level).

**Spatial autocorrelation.** When spatial autocorrelation is present, analyses conducted need to be able to adequately incorporate it [25]. While there are multiple indices and tests available to detect the presence of spatial autocorrelation, in this study we use Moran's *I*, an inferential statistic used to measure spatial autocorrelation [26], in two ways. First, to assess if the presence of spatial autocorrelation varies among different aggregations of raw data. Second, to check if the model residuals have any remaining spatial autocorrelation. Spatial models are fitted using a spatially structured error component to accommodate the spatial autocorrelation in the observed data. However, there could be well fitting models that provide good predictions but still have spatial autocorrelation indicated by significant Moran's *I* on model residuals [27].

**Bayesian spatial model: BYM model.** The observed cancer counts from different small area aggregations are modelled using a very well-known Bayesian spatial modelling framework, a BYM model [15]. The BYM model is a type of generic three level Bayesian hierarchical model, where at the first stage the observed disease counts are modelled using an appropriate likelihood, the second stage models the spatial association by specifying appropriate structures and the third stage specifies the hyperprior distributions [1].

The BYM model incorporates extra variation at the second stage via two random effect terms: a) a spatially structured component and b) a spatially unstructured component. Inclusion of both spatially structured and unstructured random effects enable the disease rates to be smoothed at both global and local level [28]. For the spatially structured term, the most commonly used approach for areal spatial data is the Gaussian Markov random field (GMRF) model which conditions area $i$ within its neighbourhood [29, 30], particularly in the form of the conditional autoregressive (CAR) prior. The detailed BYM model used is presented below:

Let $Y_i$ be the observed lung cancer counts, which can be expressed under the general disease mapping framework using a Poisson likelihood [1]:

$$Y_i \sim \text{Poisson}(E_i e^{\mu_i}), \text{ for } i = 1, 2, \cdots, N \text{areas},$$

where $E_i$ is the expected cancer counts in each area i. The present study will have different values of $N$ for different boundary levels (Table 1). The log-relative risk, denoted $\mu_i$, is often expressed as a regression equation:

$$\mu_i = \alpha + \mathbf{x}_i^T \boldsymbol{\beta} + \psi_i,$$

where the intercept $\alpha$ denotes an overall fixed effect, $\boldsymbol{\beta}$ is the covariate effects associated with the vector of covariates $\mathbf{x}_i$ relating to area $i$ and $\psi_i$ are the spatial random effects. In the context of this study, $\mathbf{x}_i$ represents the SEIFA IRSD quintiles for each area, where the first quintile (most disadvantaged) is considered as the reference category and $\boldsymbol{\beta}$ is hence a $4 \times 1$ vector representing the covariate effect of SEIFA IRSD quintiles 2, 3, 4, and 5. The Gaussian priors, $N(0, \sigma_\alpha^2)$ and $N(0, \sigma_\beta^2)$ are used for $\alpha$ and $\boldsymbol{\beta}$. The value of hyperparameters $\sigma_\alpha^2$ and $\sigma_\beta^2$ are usually chosen to be a large number. In the present study, we have used the default priors and parameters provided by the CARBayes package ($\sigma_\alpha^2 = \sigma_\beta^2 = 1,000,000$) [31]. The random effect $\psi_i = u_i + v_i$, where $u_i$ is a spatial random effect a CAR prior structure and $v_i$ is the unstructured random effect. The conditional distribution of each $u_i$ can be expressed as:

$$u_i | u_{j,i \neq j} \sim N\left(\frac{\sum_j w_{ij} u_j}{\sum_j w_{ij}}, \frac{\sigma_u^2}{\sum_j w_{ij}}\right),$$

where $w_{ij}$ are the weights defining the relationship between area $i$ and its neighbours such that:

$$w_{ij} = \begin{cases} 1, & \text{if areas } i \text{ and } j \text{ are adjacent} \\ 0, & \text{otherwise.} \end{cases}$$

The prior for the unstructured random component $v_i$ is typically an independent normal distribution,

$$v_i \sim N(0, \sigma_v^2).$$

The hyperpriors placed on the variance terms $\sigma_u^2$ and $\sigma_v^2$ are Inverse-Gamma($a$, $b$) with hyperparameters $a = 1$ (shape) and $b = 0.01$ (scale) [31].

**Implementation.** All the analyses in this study were implemented using R [32]. In total, ten BYM models were fitted covering every combination of aggregation level (mesh block, SA1, SA2, SA3 and SA4 level) and model specification (with and without covariates). Fully Bayesian inference was conducted via Markov chain Monte Carlo in the R package CARBayes version 5.2.5 [31]. For a single chain, we used 1,500,000 total iterations, discarding the first 500,000 as burnin. The posterior draws were thinned by 100 to reduce autocorrelation, leaving 10,000 iterations. Model convergence was checked through Geweke

diagnostics [33]. The spatial autocorrelation of the observed data and model residuals were checked using Moran's *I* with the R package `spdep version 1.2-8` [34].

The associated R code for model implementation and calculation of spatial autocorrelation indices are available in the Github repository: https://github.com/Farzana-Jahan/MAUP.git.

## Results

The simulated lung cancer data had a total of 20,700 lung cancer cases, with the median number per area ranging from 0.27 in meshblocks to 972 in SA4s (Table 1). There was large variation in observed and modelled SIRs across the state, with more remote western regions typically having higher rates, and the same area could have very different raw and modelled SIRs at different levels of aggregation (Figs 2–4). High spatial autocorrelation was present in the observed data at the mesh block, SA1 and SA2 levels, but not higher levels of aggregation (Table 2). After modelling, regardless of whether the socioeconomic variable was included, the residuals still showed high positive spatial autocorrelation (Table 2), but this likely could be attenuated by including additional covariates. However, the models for each level of aggregation had predominately spatially structured smoothing occurring, as demonstrated through the fraction of spatial variation, and this was slightly attenuated after including socioeconomic details for SA1 to SA3, and markedly for SA4 (Tables 3 and 4).

Differences were found in the estimates for socioeconomic coefficients depending on the level of aggregation. For instance, $\beta_4$ (least disadvantaged) was essentially 0 and non-significant at the mesh block level, while all other aggregations indicated significantly negative values (Table 4). This meant that for SA1 and above, least disadvantaged areas had lower lung cancer diagnosis rates compared to most disadvantaged areas (the reference category). In contrast, estimates for $\beta_2$ and $\beta_3$ had negative associations at higher resolution that turned to insignificance for SA3 and SA4s (Table 4). These differences could be a result of the MAUP. However, apart from meshblocks, each level of aggregation showed a consistent socioeconomic gradient from the modelled results, with increasingly negative coefficients as socioeconomic disadvantage diminished.

It is evident from the posterior summaries that inference changes with different aggregation of areas across Queensland. For models both with and without covariates, the width of the credible intervals for regression parameters ($\alpha$, $\beta_i$), and variance parameters ($\sigma_u^2$ and $\sigma_v^2$), were smaller for lower levels of aggregation (SA1 and SA2) and wider for the larger boundaries (SA3 and SA4) (Tables 3 and 4 and Figs 5 and 6), except for inference using the mesh block level data.

## Discussion and conclusion

The present study sought to understand the impact of the MAUP in spatial analysis of sparse disease data across different administrative boundary levels when using a popular Bayesian spatial model in a geographically challenging environment. We found important differences in all aspects: spatial patterns, inference from estimates and coefficient results.

One proposed method to minimising the MAUP is through using high spatial resolution [11]. However, we found that meshblocks, which typically represent 30 to 60 dwellings [35], were too sparse for a condition as rare as cancer. Estimates for model parameters at the mesh-block level often greatly differed from all other aggregation levels, and the large number of areas made it computationally intensive. While the next aggregation level of SA1s performed better, it is extremely difficult to obtain health data with location either provided as an SA1, or with the detailed street address needed to assign an SA1. Given SA2s tend to be the lowest realistic area size possible in health datasets, the large differences in modelled covariate coefficients

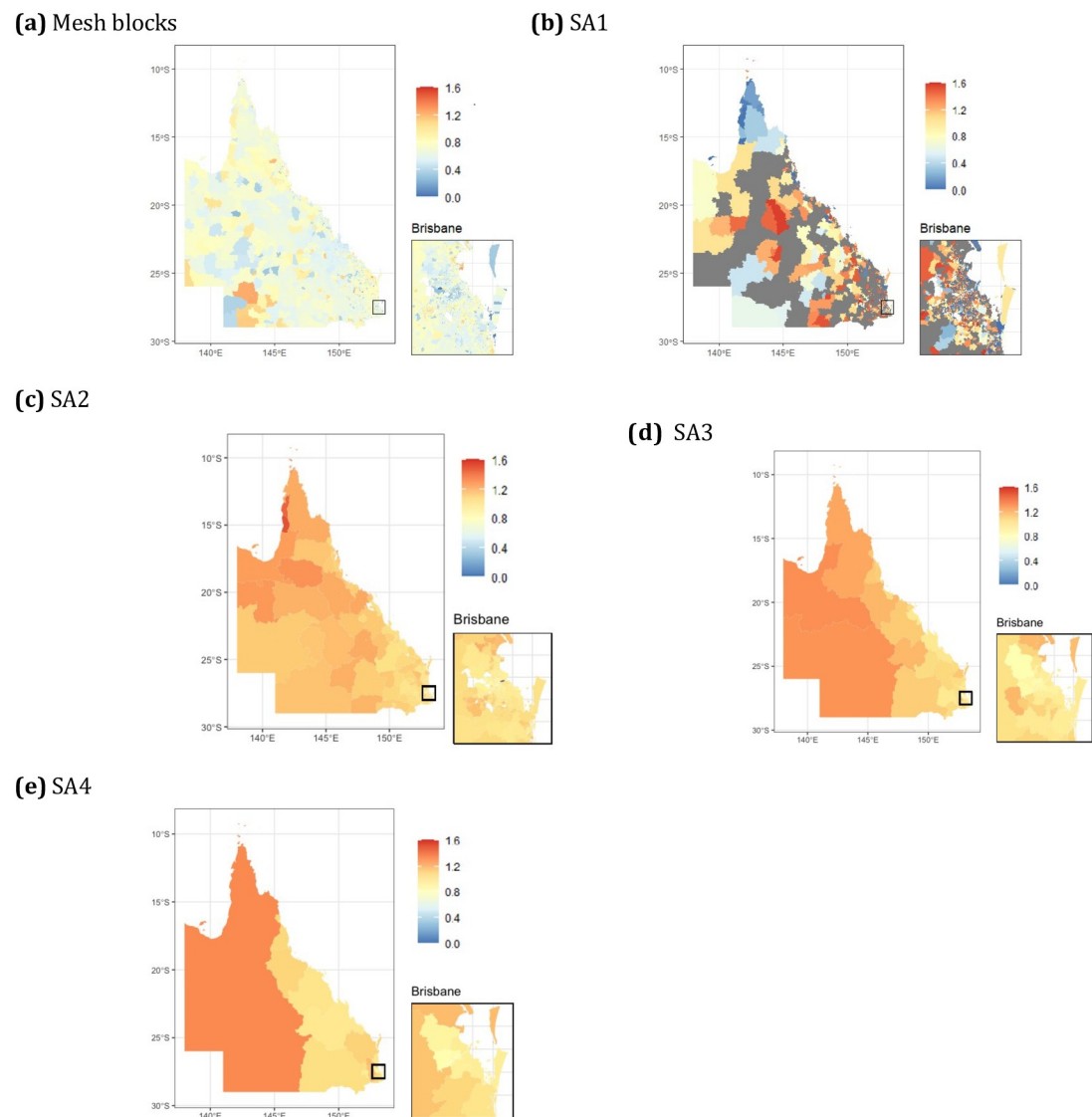

**Fig 3. Choropleth maps displaying the fitted SIR (without covariate model) from the simulated lung cancer data at multiple aggregation levels for Queensland, Australia.**

between SA1s and SA2s and different spatial patterns amplifies the importance of acknowledging the MAUP when conducting spatial analyses.

While modeling at the mesh block level can increase the sample size (i.e., the number of areas) and reduce heterogeneity within areas, it can also lead to situations where many areas have a small or zero number of cases or population, resulting in a loss of information and the potential for oversmoothing. This aligns with the findings in Fontanet et al. [16] where the smallest level failed to provide meaningful inference due to smaller populations and counts. Kok et al. [11] also found this to be the case when analysing hospitalisation rates for foot-related issues among the Indigenous population of Australia. Our findings, in conjunction with theirs, suggest that while aiming for the highest level of resolution may be optimal to reduce the biases of the MAUP, using mesh blocks can introduce other problems such as small

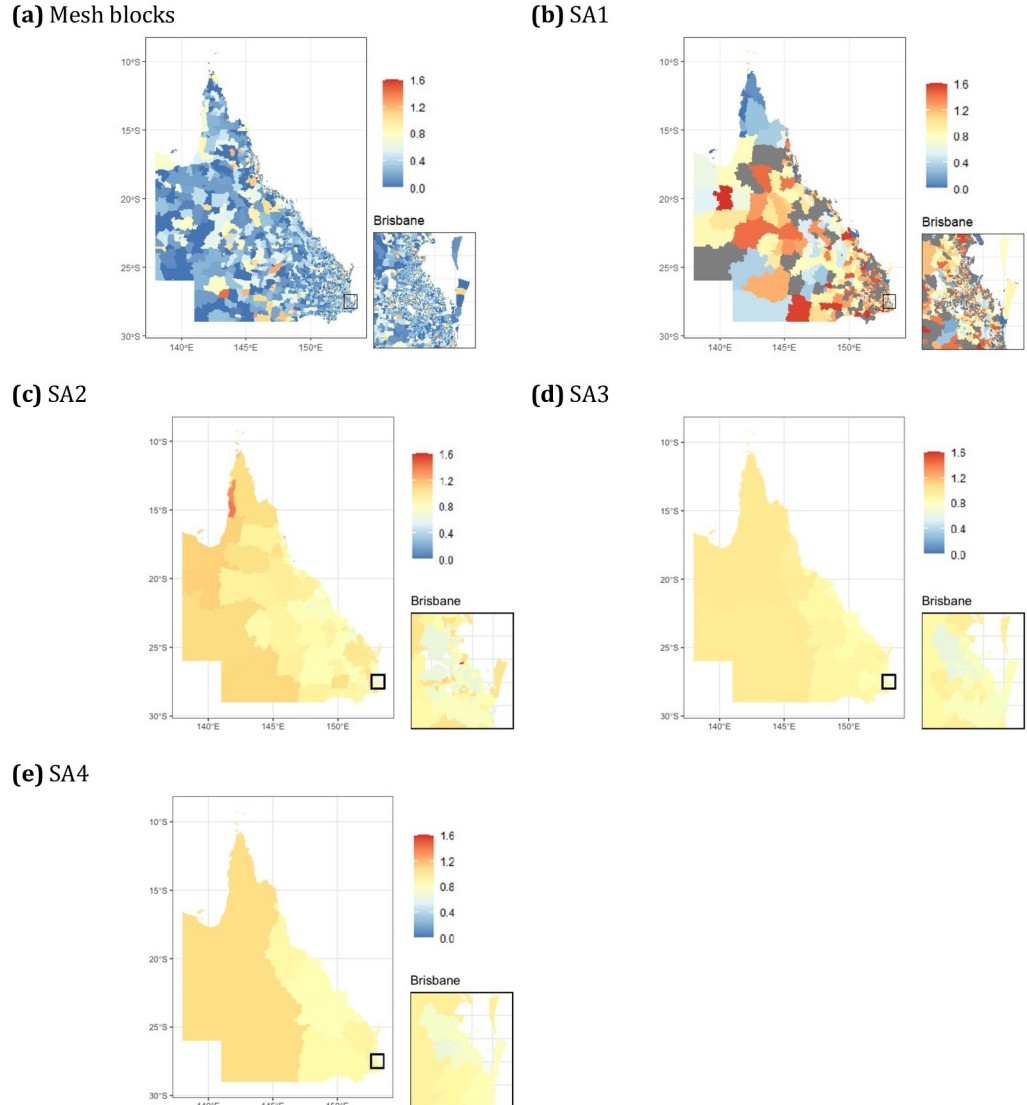

**Fig 4. Choropleth maps displaying the fitted SIR (with covariate model) from the simulated lung cancer data at multiple aggregation levels for Queensland, Australia.**

numbers, zero populations, and computational inefficiencies. In small-area health studies, particularly those involving rare diseases or small populations, sparse data can lead to highly imprecise rate estimates. This imprecision is amplified by the MAUP, causing substantial variability in the identification of disease clusters based on different spatial units [4, 36]. As a

**Table 2. Moran's *I* of observed counts and modelled residuals.**

| Aggregate level | Observed counts | | Residuals (without covariate) | | Residuals (with covariate) | |
|---|---|---|---|---|---|---|
| | statistic | p-value | statistic | p-value | statistic | p-value |
| Mesh | 0.274 | <0.0001 | 0.226 | 0.001 | 0.273 | 0.001 |
| SA1 | 0.293 | 0.001 | 0.269 | 0.0001 | 0.21 | <0.0001 |
| SA2 | 0.228 | 0.001 | 0.245 | 0.0001 | 0.216 | <0.0001 |
| SA3 | 0.022 | 0.318 | 0.062 | 0.1476 | 0.135 | 0.0237 |
| SA4 | 0.152 | 0.062 | -0.078 | 0.5518 | -0.231 | 0.917 |

**Table 3. Posterior summary of simulated lung cancer BYM model without covariates.**

| Aggregate level | Parameter estimates | | | Fraction of spatial variation |
|---|---|---|---|---|
| | $\alpha$ Median (95% CI) | $\sigma_u^2$ Median (95% CI) | $\sigma_v^2$ Median (95% CI) | |
| Mesh | -0.45 (-0.47,-0.43) | 0.08 (0.07,0.09) | 0.0013 (0.0007,0.0023) | 0.98 |
| SA1 | -0.0037 (-0.0178,0.0103) | 0.0054 (0.0033,0.0083) | 0.0009 (0.0006,0.0015) | 0.85 |
| SA2 | -0.0052 (-0.02,0.0093) | 0.007 (0.0032,0.0124) | 0.0031 (0.0014,0.0058) | 0.69 |
| SA3 | -0.0005 (-0.017,0.0164) | 0.0133 (0.0043,0.028) | 0.0048 (0.0018,0.0097) | 0.73 |
| SA4 | -0.0018 (-0.0222,0.0198) | 0.0452 (0.003,0.0633) | 0.019 (0.0028,0.0247) | 0.70 |

**Table 4. Posterior summary of simulated lung cancer BYM model with covariates.**

| Aggregate level | Parameter estimates | | | | | | | Fraction of spatial variation |
|---|---|---|---|---|---|---|---|---|
| | $\alpha$ Median (95% CI) | $\beta_1$ Median (95% CI) | $\beta_2$ Median (95% CI) | $\beta_3$ Median (95% CI) | $\beta_4$ Median (95% CI) | $\sigma_u^2$ Median (95% CI) | $\sigma_v^2$ Median (95% CI) | |
| Mesh | -0.33 (-0.55, -0.10) | -1.71 (-1.95,-1.47) | -0.471 (-0.83,-0.39) | -0.22 (-0.44,-0.01) | 0.02 (-0.20, 0.24) | 0.26 (0.23, 0.28) | 0.001 (0.0007,0.0021) | 0.99 |
| SA1 | 0.12 (0.09,0.15) | -0.07 (-0.11, -0.03) | -0.12 (-0.17, -0.08) | -0.18 (-0.24,-0.15) | -0.28 (-0.32,-0.22) | 0.002 (0.001,0.003) | 0.0009 (0.006,0.0015) | 0.69 |
| SA2 | 0.199 (0.17,0.23) | -0.131 (-0.17,-0.09) | -0.229 (-0.27,-0.19) | -0.287 (-0.33,-0.24) | -0.462 (-0.51,-0.41) | 0.0015 (0.0008,0.003) | 0.0012 (0.0007,0.0019) | 0.55 |
| SA3 | 0.10 (0.0598,0.147) | -0.047 (-0.11,0.017) | -0.06 (-0.13,0.005) | -0.153 (-0.23,-0.08) | -0.27 (-0.34,-0.2) | 0.004 (0.002,0.009) | 0.003 (0.001,0.005) | 0.57 |
| SA4 | 0.13 (0.05,0.22) | -0.08 (-0.21,0.04) | -0.14 (-0.285,0.008) | -0.19 (-0.33,-0.06) | -0.32 (-0.46,-0.16) | 0.008 (0.002,0.024) | 0.005 (0.002,0.012) | 0.32 |

result, health interventions and resource allocations might be misdirected due to the unreliability of the spatial analysis.

In addition, advantages of using the well-defined boundary levels of SA1 and SA2 also enables easy incorporation of additional demographic and socio-economic data into disease models [37]. Detailed spatial data at these levels can often be more effectively communicated to stakeholders, policymakers, and the public, facilitating better understanding and engagement.

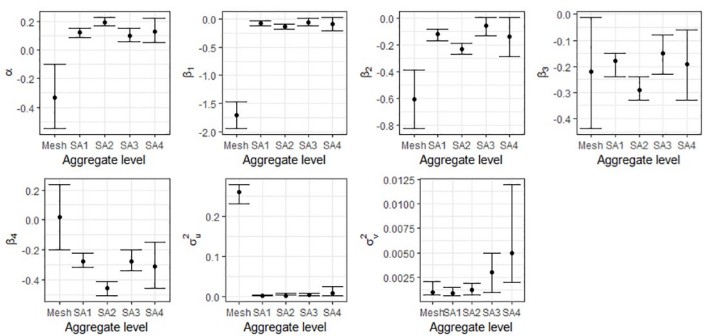

**Fig 5. Credible intervals of parameters for BYM model with covariate.**

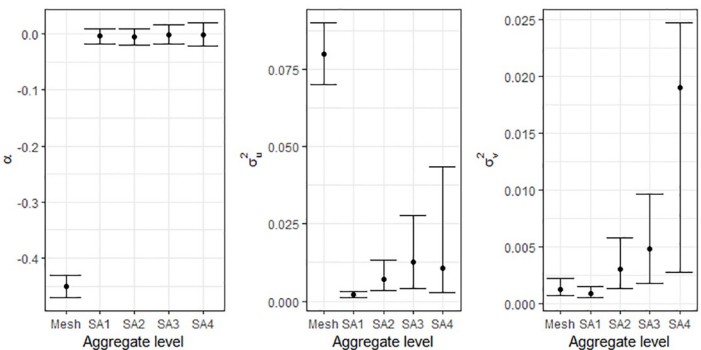

**Fig 6. Credible intervals of parameters for BYM model without covariate.**

With higher aggregations, we found that the spatial autocorrelation of the data is no longer significant (for SA3 and SA4 levels), so use of spatial modelling/smoothing may not be required. But since SA1 and SA2 level data has significant spatial autocorrelation, analysis of cancer incidence data at these levels should be performed using a model that accounts for spatial autocorrelation.

The study has several limitations arising from data and methodology restrictions. Data constraints required SA1 and meshblock level data to be created from the SA2 level. The high level of heterogeneity within larger areas such as SA3s and above means that area-level socioeconomic variables are not released, and this may have impacted results, despite using recommended methodology to create these. However, this is also a feature of the MAUP: both the outcome and its covariates are impacted by aggregation. Some models did not fit well, considering the large amount of spatial correlation remaining in the residuals. As inadequate adjustment for spatial autocorrelation can bias fixed effects, in practical applications researchers are encouraged to explore different spatial structures and more useful covariates, particularly spatially correlated covariates. Our focus was on the commonly used Australian areal structures (Mesh block, SA1, SA2, SA3 and SA4), but rezoning boundaries (while keeping land areas consistent) can also identify the MAUP. We were interested in seeing how sparse data performed in the challenging Australian context, but our approach could be repeated considering other health conditions of varying prevalence, or including all of Australia.

In conclusion, as a result of this work we propose an alternative narrative regarding the optimal aggregation level for spatial modelling. Although it is common to analyse spatial data at the lowest level of aggregation possible, we suggest that a natural balance is instead preferable for inference, modelling and most importantly, reducing the impact of the MAUP. In the context of Australia, we found that although analysing cancer incidence at the mesh block level (lowest aggregation) alleviates the concerns of the MAUP, is not advisable due to the high computational burden and extremely noisy and sparse data. Moreover, when using higher aggregation levels such as SA3 and SA4, uncertainty increases, making it more difficult to identify useful covariates as a result of the MAUP. Thus, for the lung cancer data used in this work, we recommend modelling at the SA1 or SA2 level. Due to the sparsity inherent in disaggregating cancer data, we hypothesise that a similar conclusion may be drawn for other cancers and other countries with areas of low population density.

## Supporting information

**S1 Data.**
(XLSX)

## Author Contributions

**Conceptualization:** Farzana Jahan, Aiden Price, Helen Thompson, Jessica Cameron, Susanna M. Cramb.

**Data curation:** Jessica Cameron, Susanna M. Cramb.

**Formal analysis:** Farzana Jahan, Shovanur Haque.

**Funding acquisition:** Farzana Jahan, Aiden Price, Helen Thompson, Jessica Cameron, Susanna M. Cramb.

**Investigation:** Farzana Jahan, Conor Hassan, Wala Areed, Jessica Cameron.

**Methodology:** Farzana Jahan, Aiden Price, Helen Thompson, Jessica Cameron, Susanna M. Cramb.

**Project administration:** Susanna M. Cramb.

**Software:** Farzana Jahan, Shovanur Haque, James Hogg.

**Supervision:** Farzana Jahan, Jessica Cameron, Susanna M. Cramb.

**Validation:** Farzana Jahan, James Hogg.

**Visualization:** Farzana Jahan, Shovanur Haque.

**Writing – original draft:** Farzana Jahan, Shovanur Haque.

**Writing – review & editing:** Farzana Jahan, James Hogg, Aiden Price, Conor Hassan, Wala Areed, Helen Thompson, Jessica Cameron, Susanna M. Cramb.

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
