## [Decision Letter · Decision Letter 0]

27 May 2024

PONE-D-24-13684Assessing the influence of the modifiable areal unit problem on Bayesian disease mapping in Queensland, AustraliaPLOS ONE

Dear Dr. Jahan,

Thank you for submitting your manuscript to PLOS ONE. After careful consideration, we feel that it has merit but does not fully meet PLOS ONE’s publication criteria as it currently stands. Therefore, we invite you to submit a revised version of the manuscript that addresses the points raised during the review process.

**ACADEMIC EDITOR: ****Kindly revise the research paper as per the reviewer comments and resubmit it soon. ** 

We look forward to receiving your revised manuscript.

Kind regards,

T. Ganesh Kumar, PhD

Academic Editor

PLOS ONE

“QUT Centre for Data Science.”

“Ethical Approval

Only simulated data were used in this manuscript, data was simulated by Cancer

Council Queensland following the appropriate ethical consideration. This was

approved by the Data Custodians of the original data.

Funding

This research was funded by QUT Centre for Data Science.

Data availability

Datasets used in the research are not publicly available. However, the data can be

made available on request to the authors.”

“QUT Centre for Data Science.”

5. In the online submission form you indicate that your data is not available for proprietary reasons and have provided a contact point for accessing this data. Please note that your current contact point is a co-author on this manuscript. According to our Data Policy, the contact point must not be an author on the manuscript and must be an institutional contact, ideally not an individual. Please revise your data statement to a non-author institutional point of contact, such as a data access or ethics committee, and send this to us via return email. Please also include contact information for the third party organization, and please include the full citation of where the data can be found.

7. We note that Figures 1, 2, 3, and 4 in your submission contain [map/satellite] images which may be copyrighted. All PLOS content is published under the Creative Commons Attribution License (CC BY 4.0), which means that the manuscript, images, and Supporting Information files will be freely available online, and any third party is permitted to access, download, copy, distribute, and use these materials in any way, even commercially, with proper attribution. For these reasons, we cannot publish previously copyrighted maps or satellite images created using proprietary data, such as Google software (Google Maps, Street View, and Earth). For more information, see our copyright guidelines: http://journals.plos.org/plosone/s/licenses-and-copyright.

1. You may seek permission from the original copyright holder of Figures 1, 2, 3, and 4 to publish the content specifically under the CC BY 4.0 license. 

Additional Editor Comments:

Dear Authors,

Your paper is quality for publication. We have received the comments from the reviewers. Both reviewers are accepted the research paper with few comments.

The authors needs to revise the manuscript as per the reviewer comments. Kindly revise your paper as per the comments and resubmit it soon.

Reviewers' comments:

Reviewer's Responses to Questions

**Comments to the Author**

1. Is the manuscript technically sound, and do the data support the conclusions?

Reviewer #1: Yes

Reviewer #2: Yes

2. Has the statistical analysis been performed appropriately and rigorously? 

Reviewer #1: Yes

Reviewer #2: Yes

3. Have the authors made all data underlying the findings in their manuscript fully available?

Reviewer #1: No

Reviewer #2: Yes

4. Is the manuscript presented in an intelligible fashion and written in standard English?

Reviewer #1: Yes

Reviewer #2: Yes

5. Review Comments to the Author

Reviewer #1: The manuscript provides a comprehensive introduction to the modifiable areal unit problem (MAUP) and its significance in spatial epidemiology, particularly in Bayesian disease mapping. The literature review is thorough, referencing key studies that highlight the challenges and previous approaches to addressing MAUP. The methodology section is well-detailed, describing the Bayesian disease mapping techniques used, including the choice of spatial units and statistical models. The manuscript justifies the selection of spatial units, explaining how different aggregations might influence the results. Advanced statistical methods are employed to account for spatial autocorrelation and to model disease risk, demonstrating a rigorous approach. Results are presented clearly, with appropriate use of tables and maps summaries to illustrate the impact of MAUP on disease mapping. Statistical analyses are robust, showing how different spatial aggregations affect the estimates of disease risk. The results are consistent with the hypotheses, demonstrating the influence of MAUP on the findings. The discussion effectively links the results back to the research questions and the broader literature on MAUP and disease mapping. Conclusions are logically derived from the results, emphasizing the importance of considering MAUP in spatial epidemiology. Recommendations for future research and policy implications are clearly stated, highlighting the practical significance of the findings. The manuscript critically assesses the impact of MAUP, discussing potential solutions and areas for further investigation. The data used are directly relevant to the study's objectives, focusing on disease incidence in Queensland, Australia. The spatial and temporal resolution of the data is appropriate for examining the effects of MAUP on Bayesian disease mapping. The analyses are thorough, using multiple spatial aggregations to demonstrate how MAUP affects disease risk estimates. Statistical techniques are appropriately applied, ensuring that the findings are robust and reliable. The interpretation of the results is sound, clearly showing how different spatial units lead to variations in disease risk estimates. The manuscript discusses the implications of these variations, supporting the need for careful consideration of MAUP in spatial epidemiological studies. The data support the conclusions drawn, with clear evidence that MAUP significantly influences Bayesian disease mapping outcomes. The manuscript provides a strong case for the importance of addressing MAUP, backed by rigorous statistical analysis and comprehensive data interpretation. Overall, the manuscript "Assessing the influence of the modifiable areal unit problem on Bayesian disease mapping in Queensland, Australia" is technically sound. The data are credible and relevant, and the analyses are thorough and appropriately applied. The conclusions are well-supported by the data, providing valuable insights into the impact of MAUP on disease mapping and highlighting the need for careful spatial analysis in epidemiological studies.

Reviewer #2: 1. Why is it significant to investigate the impact of the MAUP in regions with sparse data, particularly in countries like Australia with less populated areas?

2. How MAUP affects the analysis of spatially aggregated data? Why it is a concern in spatial analysis?

3. What advantages do area structures with moderate resolution offer in terms of dealing with the MAUP?

4. What types of Bayesian spatial models were employed in the study, and how do they account for spatial correlation and variability?

5. What are the potential benefits of using SA1 or SA2 levels for spatial modelling according to the authors?

6. PLOS authors have the option to publish the peer review history of their article (what does this mean?). If published, this will include your full peer review and any attached files.

Reviewer #1: **Yes: **Dr.M.Subbulakshmi

Reviewer #2: No

---

## [Author Response · Author response to Decision Letter 0]

25 Aug 2024

Added the response to reviewer comments document and added the response to journal requirements in the cover letter to the editor.

---

## [Decision Letter · Decision Letter 1]

18 Oct 2024

Assessing the influence of the modifiable areal unit problem on Bayesian disease mapping in Queensland, Australia

PONE-D-24-13684R1

Dear Dr. Jahan,

We’re pleased to inform you that your manuscript has been judged scientifically suitable for publication and will be formally accepted for publication once it meets all outstanding technical requirements.

Kind regards,

T. Ganesh Kumar, PhD

Academic Editor

PLOS ONE

Additional Editor Comments (optional):

The authors have answered the reviewer's comments and updated the contents in the revised versions.

I accept the paper for the further process.

Reviewers' comments:

Reviewer's Responses to Questions

**Comments to the Author**

1. If the authors have adequately addressed your comments raised in a previous round of review and you feel that this manuscript is now acceptable for publication, you may indicate that here to bypass the “Comments to the Author” section, enter your conflict of interest statement in the “Confidential to Editor” section, and submit your "Accept" recommendation.

Reviewer #1: All comments have been addressed

Reviewer #2: All comments have been addressed

2. Is the manuscript technically sound, and do the data support the conclusions?

Reviewer #1: Yes

Reviewer #2: Yes

3. Has the statistical analysis been performed appropriately and rigorously? 

Reviewer #1: Yes

Reviewer #2: Yes

4. Have the authors made all data underlying the findings in their manuscript fully available?

Reviewer #1: Yes

Reviewer #2: Yes

5. Is the manuscript presented in an intelligible fashion and written in standard English?

Reviewer #1: Yes

Reviewer #2: Yes

6. Review Comments to the Author

Reviewer #1: The manuscript "Assessing the influence of the modifiable areal unit problem on Bayesian disease mapping in Queensland, Australia" presents to assess different geographical regions' vulnerability to the MAUP when data are relatively sparse to inform researchers' choice of aggregation level for fitting spatial models. The authors have demonstrated a commendable effort in developing a system for this purpose. Providing clear headings, subheadings, and transitions between sections would improve the flow of information and aid in understanding the research methodology and findings. Providing a comprehensive overview of the algorithms, parameters, and preprocessing techniques used would enhance the reproducibility and understanding of the study. The manuscript should adhere to Standard English conventions to ensure clarity and coherence. This includes using appropriate grammar, punctuation, and sentence structure throughout the text. Ensure consistency in terminology and terminology usage throughout the manuscript. Additionally, maintaining cohesion between sections and paragraphs by clearly establishing logical connections between ideas would enhance readability and comprehension. Additionally, complex technical terms and concepts should be explained clearly to accommodate readers with varying levels of expertise in the subject matter. The introduction provides a comprehensive background on the Modifiable Areal Unit Problem (MAUP) and its significance in Bayesian disease mapping. However, the review of literature could be expanded to include more recent studies or alternative methodologies that address MAUP. The specific relevance of Queensland, Australia, to the study is well articulated. It would be helpful to provide more context on why Queensland was chosen and how its geographic and demographic characteristics might influence the findings. The methodology section is robust, but it would benefit from a more detailed explanation of the Bayesian models used and how they were adjusted to account for MAUP. The sources of data and their quality are crucial for the validity of the results. More information on the data collection process and any limitations of the datasets used would strengthen this section. The statistical techniques employed for assessing the impact of MAUP on disease mapping are sound, but additional details on the assumptions made and how they were validated would be useful. The results are presented clearly, but the discussion could be enhanced by providing more insight into how the findings compare with previous studies on MAUP and Bayesian disease mapping. The discussion section does a good job of outlining the implications of the findings for Bayesian disease mapping. However, it could benefit from a deeper exploration of how the identified issues with MAUP could influence public health decision-making and policy in Queensland. Ensure that all references are up-to-date and relevant. Adding recent publications or key studies related to MAUP and Bayesian disease mapping could enhance the credibility and depth of the literature review. Addressing these considerations will enhance the intelligibility and readability of the manuscript, thereby improving its overall impact and effectiveness in communicating the research findings to the scientific community. Overall, while the manuscript presents a promising approach to Assessing the influence of the modifiable areal unit problem on Bayesian disease mapping in Queensland, Australia strengthen the technical soundness of the study and the support for its conclusions.

Reviewer #2: The raised queries are answered and modified in the revised manuscript in few areas. It can be accepted.

7. PLOS authors have the option to publish the peer review history of their article (what does this mean?). If published, this will include your full peer review and any attached files.

Reviewer #1: No

Reviewer #2: No

---

## [Editor Report · Acceptance letter]

31 Oct 2024

PONE-D-24-13684R1 

PLOS ONE

Dear Dr. Jahan, 

I'm pleased to inform you that your manuscript has been deemed suitable for publication in PLOS ONE. Congratulations! Your manuscript is now being handed over to our production team.

Kind regards, 

on behalf of

Dr. T. Ganesh Kumar 

Academic Editor

PLOS ONE